



# The roles of surface processes on porphyry copper deposits preservation

Beatriz Hadler Boggiani[1], Tristan Salles[1], Claire Mallard[1], and Nicholas Atwood[2]

[1]School of Geosciences, The University of Sydney, Sydney, Australia
[2]BHP Exploration, Tucson, Arizona, USA

**Correspondence:** Beatriz Hadler Boggiani (beatriz.boggiani@sydney.edu.au)

**Abstract.** Porphyry copper deposits typically originate within subduction zones at 2 to 5 km depths. These deposits are exhumed due to the influence of tectonic forces and climate-driven erosion. Porphyry copper deposits are currently only mineable at relatively shallow depths, and their prospectivity relies on a balance between the rate of exhumation and preservation. In this study, we evaluate the impact of surface processes on the preservation or exhumation of porphyry copper deposits. To do so, we rely on a global-scale numerical model (*goSPL*), which simulates landscape dynamics and associated erosion and deposition patterns over geological time scales. High-resolution Cenozoic simulations incorporate published open-source global paleo-climate and paleo-elevation datasets, and have been fine-tuned using contemporary data. We then calculate exhumation rates by comparing the ages of known porphyry copper deposits and their simulated emplacement depths based on modelled erosion-deposition values. Obtained average exhumation rates vary from $10^{-2}$ to $10^{-1}$ km/Myr, with an overall difference of 0.04 mm/yr when compared to independent erosion rate estimates available from published studies. The predicted global mean emplacement depths range from 1 to 3 km. To highlight the influence of paleo-reconstructions on exhumation rate estimates, we analyse simulated erosion rates across the Andean region using two distinct paleo-climate models and find significant spatial and temporal differences across the Central Andes. While our landscape evolution model successfully predicts the known emplacement depths for the North and South Andean deposits younger than 20 Myr, it also predicts depths exceeding 6 km for Central Andean deposits older than 60 Myr. We attribute these mismatches to a combination of limitations related to model assumptions and inputs resolutions. Our results show the intricate connection between deposit preservation and surface processes. Our method offers an addition to the traditional porphyry copper exploration toolkit that link geological observations to plate tectonics dynamics and paleo-climatic reconstructions.

## 1 Introduction

The association between porphyry copper deposits (PCD) and alkaline/calc-alkaline magmatic arcs has been well established for at least 50 years (Sillitoe, 1972). These deposits most commonly originate from the combination of: (i) rapid convergence rates (around 100 km/Myr) (Butterworth et al., 2016), (ii) subduction obliquity (around 15°) (Butterworth et al., 2016), (iii)





subducting plate age (between 25-70 Myr) (Butterworth et al., 2016; Sillitoe, 2010), (iv) distance from the subducting trench

boundary (>2000km) (Butterworth et al., 2016), and (v) a thick crust (> 45 km) (Lee and Tang, 2020).

Assuming the aforementioned conditions are fulfilled, the porphyry bodies are emplaced at a certain depth, with the emplacement depth (ED) referring to the depth at which magmatic fluids encounter a favorable environment within the Earth's crust (Lee and Tang, 2020; Sillitoe, 2010), characterized by optimal pressure, temperature, and salinity conditions (Richards, 2022; Sillitoe, 2010). Predictions regarding the depth of emplacement are inferred from fluid inclusion analysis or age temperature

data (McInnes et al., 2005; Landtwing et al., 2005; Noury and Calmus, 2021) and typically range between 1 to 5 km, with lower magmatic temperatures typically associated with shallower intrusions, while higher temperatures correspond to greater depths (Richards, 2022). It is common for economic porphyry deposits to have emplacement depths between 4-5 km (Zhao et al., 2022). Exceptional cases exist where deposits have been emplaced at depths greater than 5 km. For example, the Middle Paleocene (60.5 Ma) Butte body in Montana (USA) is believed to have been emplaced at an unusual depth between 5 and 9 km

(Rusk et al., 2008). On the other hand, other deposits such as the 3.5 Ma Panguna in Papua New Guinea have been emplaced between 2 and 3 km (Eastoe, 1978; Cooke et al., 2005).

The exhumation trajectory and preservation of porphyry copper deposits are primarily influenced by climate-driven erosion and tectonic uplift. Exhumation rates (ER) are strongly tied to precipitation history, substrate erodibility, slope angles and direction of propagation of the deformation (Stalder et al., 2020). Whilst ER between 0.1 and 1.0 km/Myr are considered typical for a

moderately active mountain belt (Montgomery and Brandon, 2002), these rates can vary significantly at regional scale due to tectonic and climatic forces. For example, hyper-arid climates such as the Central Andes (Stern, 2004) have ER that do not exceed 0.25 km/Myr (Stalder et al., 2020). Comparatively, regions such as Papua New Guinea, known for their young PCDs, elevated precipitation rates, and rapid plate convergence creating high uplift rates, exhibit some of the highest ER, greater than 10 km/Myr (Baldwin et al., 2004).

Yanites and Kesler (2015) calculated the exhumation rates of 314 Cenozoic PCDs distributed globally and reported values ranging from 0.03 to 1.4 km/Myr by assuming uniform emplacement depth of 2 km for all their PCDs and extrapolating present-day climatic conditions for the entire Cenozoic. This reveals a pattern where younger deposits in wetter regions experienced rapid exhumation, while older deposits are more prevalent in arid regions. Apart from Yanites and Kesler (2015) and despite the critical role that climate plays on landscape evolution, its implication for PCDs preservation has been overlooked

in recent studies (Butterworth et al., 2016; Diaz-Rodriguez et al., 2021). Butterworth et al. (2016) and Diaz-Rodriguez et al. (2021), employed machine learning algorithms to predict the prospectivity of porphyry copper districts in the Americas. Both studies largely ignored the impact of climate and paleotopographic variations on PCD exhumation and preservation. This study challenges the assumption of singular emplacement depth as well as static climatic and paleotopographic conditions over geological time by using a series of time evolving databases (Valdes et al., 2020; Li et al., 2022; Stern, 2004) over the past 65 Myr

(see section 3.2).

Here, we leverage global-scale surface process simulations to estimate global rates of exhumation and emplacement depths of known PCDs, and assess their preservation prospects. To do so, we use the refined paleo-elevation dataset produced in Salles et al. (2023), applying *goSPL* model (Salles et al., 2020) and estimate exhumation rates and emplacement depths of





the same 314 deposits presented in Yanites and Kesler (2015). We then compare exhumation rate and emplacement depth

values obtained from the landscape evolution model with data obtained via independent published methods (*e.g.* fission track

analysis). We explore discrepancies and their potential causes, such as factors related to the paleo-reconstructions or limitations

inherent to the numerical approach.

## 2 Global paleo-elevation reconstruction

### 2.1 Landscape evolution model

To estimate exhumation rates and emplacement depths at global scale, we use *goSPL* (Salles et al., 2020), an open-source

scalable parallel numerical model. *goSPL* accounts for river incision and soil creep, processes which are considered to be the

main drivers of landscape changes over geological time scale (Bonetti and Porporato, 2017). Additional capabilities (*e.g.*, multi-

lithology sediment tracking, deposition in marine environments, sediment compaction, stratigraphic record) are not considered

in this study and we evaluate the impact of surface processes on landscape dynamics (Salles et al., 2023) by imposing a series

of spatially and temporally varying topographic and climatic histories.

In *goSPL*, erosion occurring in upstream catchments is linked to basin sedimentation via a multiple flow direction algorithm

that routes both water and sediment flux towards multiple downstream nodes preventing the locking of erosion pathways along

a single direction and allowing the distribution of the corresponding flux in downstream regions. The model's main equation,

the continuity of mass, has the following common form:

$$\frac{\partial z}{\partial t} = U + \kappa \nabla^2 z + \epsilon P^d (PA)^m \nabla z^n \qquad (1)$$

where the changes in surface elevation ($z$ in meters) with time ($t$ in years) are dependent on the tectonic forcing $U$ (m/yr); a

diffusion equation used to simulate hillslope processes, with the diffusion coefficient $\kappa$ set to 0.5 m²/yr (Salles et al., 2018);

and on the stream power law to evaluate the impact of fluvial processes by quantifying the rate of incision from local slope and

amount of water passing through a cell. We set $m$ to 0.5 and $n$ to 1 (dimensionless empirical constants, Salles et al. (2023)).

The precipitation-independent component of erodibility $\epsilon$ (set to $4.0 \times 10^{-7}$/yr, ) is constant throughout the simulation time

and is tied to the choice of $m$. The erodibility coefficient is scalable with local mean annual precipitation rate ($P$), which is a

positive exponent ($d$) obtained from field-based relationships (Murphy et al., 2016) and set to $0.42$. It is worth nothing that $d$

is incorporated in the formulation for enhancing the weathering impact of precipitation and its response in river incisions. The

upstream total area ($A$) and local precipitation ($P$) are combined to form the water flux ($PA$) (Salles et al., 2018, 2023).

To solve the flow discharge globally ($PA$), we use a parallel implicit drainage area (IDA) method (Richardson et al., 2014;

Salles et al., 2019) in a Eulerian reference frame, expressed in the form of a sparse matrix composed of diagonal terms set

to unity and off-diagonal terms corresponding to the immediate neighbours of each vertex composing the spherical mesh. A

similar technique is used to solve sediment transport and continental erosion. We use the Richardson solver with block Jacobian

preconditioning to solve the IDA algorithm using the PETSc library (Smith et al., 2012) following the approach described in





Richardson et al. (2014). Although Eq. 1 does not account for deposition in alluvial plains or along gentle slopes, it simulates continental deposition in depressions and endorheic basins.

## 2.2   Paleo-elevation and paleo-precipitation forcings

Here we use the global scale simulations from Salles et al. (2023), which are based on the PALEOMAP Paleogeographic Atlas (Scotese and Wright, 2018). The PALEOMAP paleo-elevation maps are structured regular grids with a resolution equivalent

to 10 km cell length at the Equator ($0.1° \times 0.1°$) defined at 5 Myr intervals. The paleo-elevation reconstructions are based on lithofacies and paleoenvironmental datasets, supplemented and updated with regional paleogeographic atlases (Scotese and Schettino, 2017; Scotese and Wright, 2018) over the course of more than 40 years (Ziegler et al., 1985). Our choice for the PALEOMAP atlas as the foundational dataset was driven by its unique availability as a global open-source option with a resolution ($0.1°$ or finer) appropriate for landscape evolution modelling. However, alternative datasets (e.g. Markwick and

Valdes, 2004; Vérard et al., 2015; Straume et al., 2019) could equivalently be used.

To predict erosion-deposition rates, Salles et al. (2023) rely on the paleo-precipitation simulations from HadCM3BL-M2.1aD29, which is a variant derived from the coupled atmosphere-ocean-vegetation Hadley Centre model (Valdes et al., 2020) (resolution of $3.75° \times 2.5°$) and has been built using the PALEOMAP Paleogeographic Atlas (Scotese and Wright, 2018). For each time interval, this climate model has been run for at least 5,000 model years to reach a dynamic equilibrium of the deep ocean

(Valdes et al., 2020). The climate model is established under two time-dependent boundary conditions: the solar constant and the atmospheric $CO_2$ concentrations. We use the workflow set out in Salles et al. (2023) that adopted the $CO_2$ local weighted regression curve from Foster et al. (2017) based on their choice of HadCM3 climate simulations for the paleo-precipitation maps.

The input files for *goSPL* are composed of an icosahedral mesh with more than 10 million nodes and 21 million cells. They are

generated by resampling the global temporal grids into this mesh. The average resolution of each cell is about 5 km, corresponding to $0.05°$ resolution at the equator. The design approach (Salles et al., 2023, 2020) aims to achieve dynamic equilibrium between erosion rates balanced by tectonics under a steady boundary condition (rainfall, tectonic uplift, and erodibility) (Eq. 1 and Fig. 1). For each individual time-slice, two sets of simulations were run over 168 CPU (Salles et al., 2023). From there, we estimate physiographic characteristics and associated water and sediment dynamics.

An initial simulation is performed over 2 Myr to appraise landscape evolution, water discharge, and sediment transport, using selected topographical and precipitation scenarios (Salles et al., 2023). This first simulation does not account for tectonic processes, resulting in an excessive fingerprint on the reconstructed landscape from surface processes, leading to a lowering of major long-lived orogenic belts and upland areas as well as creating extensive floodplains (Fig. 1a and b). Using a filtering approach that removes high amplitude short wavelength variations, we ensure that these morphological features are not accounted

for in the correction step, resulting in minimal changes in hypsometry ($\leq 0.5\%$, Fig. 1c). Consequently, the final elevations were corrected by assimilating paleo-elevation information. The resultant paleo-elevation model integrates the surface process features from the initial simulation, recreating more detail when compared to the initial paleo-elevation, where canyons, incised channels and basins can be identified (Fig. 1a and b). By calculating the regional differences between the paleo-elevation data



and the modified enhanced paleo-elevation, we can derive a vertical tectonic forcing map delineating uplift and subsidence
rates. The second and final round of simulations are run considering the same initial input files, however with the computed
tectonic forcing added to it (Salles et al., 2023) (Fig. 1). The simulation runs until it achieves dynamic equilibrium (i.e., erosion
rates compensate tectonic ones), which usually occur within the initial million years of landscape evolution (Fig. 1c) (Salles
et al., 2023). From the output of this second simulation, it is possible to evaluate water and sediment flux for the time-slice, as
well as catchment characteristics, such as river network, drainage areas, erosion, and deposition rates (Fig. 2 and Fig. 3a and
d).

Using the outputs from the second simulation, Eq. 1 is then calibrated at present day based on (i) estimates of suspended
sediment flux, which is equivalent to 12.8 Gt/yr (from the BQART model Syvitski and Milliman (2007); Peucker-Ehrenbrink
(2009)) and (ii) with modern estimates of average global erosion rates, (Willenbring et al., 2013) where the mean value is 63
m/Myr with a standard deviation of 15 m/Myr (Fig. 3b). The average present-day global modelled erosion rate is 73 m/Myr,
with a modelled sediment flux of 12.15 Gt/yr (assuming an average density of 2.7 g/cm$^3$) (Salles et al., 2023). Using the IDA
algorithm (Richardson et al., 2014), basin characteristics and locations for the largest sediment flux and water discharge are
extracted (Fig. 3d).

The model accurately accounts for the discharge-area, scaling relationship between water and sediment flux for the major rivers
at present-day (Whipple and Tucker, 1999). For example, the predictions from the Amazon River discharge and sediment
flux, lie well between estimates from the Land2Sea database (Peucker-Ehrenbrink, 2009) (6,591 to 7,570 km$^3$/yr) and are
∼4% below the sediment production rate when compared with cosmogenic nuclide analysis (∼610 Mt/yr) (Wittmann et al.,
2011; Salles et al., 2023). The parameterisation from present-day suspended sediment flux and global erosion rates are then
extrapolated into the past enabling the computation of temporal trends in global sediment fluxes, which can be linked to changes
in continental elevation and precipitation patterns (Salles et al., 2023) (Fig. 2a and 3). The predicted trends of sediment fluxes
produce reasonable records during the Meso-Cenozoic, and are comparable with observations for other time periods (Wilkinson
and McElroy, 2007) (Fig. 3a). Furthermore, multiple spikes in erosion flux align with major orogenic events, overlaying the
trend observed throughout the Cenozoic (Salles et al., 2023) (Fig. 2a).

## 2.3 Landscape evolution model outputs extraction

The data produced in the simulation are processed with open-source workflows provided in GitHub repositories: https://github.
com/Geodels/gospl-global-workflows and https://github.com/Geodels/gospl/ (Salles et al., 2020). All the notebooks are writ-
ten in Python Programming Language and supported by a wide range of libraries, of which the most employed are Numpy
(Oliphant, 2006), Pandas (McKinney, 2011), XArray (Hoyer and Hamman, 2017), pyGPlates (Mather et al., 2023) and PyGMT
(Uieda et al., 2021).

The output files created with *goSPL* are exported to netCDF grids containing, for each time step, the following variables: ele-
vation (m), cumulative erosion/deposition (m), water flux discharge accounting for lakes (m$^3$/yr), water flux in rivers excluding
lakes (m$^3$/yr), sediment flux in rivers (m$^3$/yr) and rainfall (m/yr).




To access the information for a desired region over time, in our case the porphyry copper deposit location and its immediate surrounding region (*i.e.* within 25 km), it is necessary to track the region's position through tectonic plate reconstruction. This is achieved using the associated tectonic plate rotation file and its geometries (PALEOMAP Paleogeographic Atlas from Scotese and Wright (2018)). By determining the region's position over time, we then extract the relevant variables at each time step.

In our scenario, we used the cumulative erosion/deposition values over time to estimate erosion rates (m/yr), which will correspond to exhumation rates (km/Myr) over longer periods of time. With this information and the predicted ages of PCDs (Singer et al., 2008), we then infer the initial depth of emplacement.

## 3 Reconstructing exhumation history for porphyry copper deposits

We conducted an analysis of 314 porphyry copper deposits distributed globally and available from Yanites and Kesler (2015). These deposits range in age from the Paleocene to the Pleistocene (66 to 1 Ma) (Fig. 4).

### 3.1 Tectonic settings and deposits

We first categorised the deposits within three geodynamic regions/settings: the Alpine-Himalayan Orogeny, the Western Cordillera/Andean Belt, and the West-Pacific Orogenic Belt.

The Western Cordillera and the Andean Belt belong to distinct subduction zones, active at different times, yet they share a similar geotectonic context, characteristic of continental margin settings (Ramos, 2009). The Western Cordillera (also known as the North American Cordillera) can be traced back to the breakup of Rodinia (Monger and Price, 2002; Clennett et al., 2020), and has been primarily shaped by the interactions between the North American plate and the surrounding oceanic lithospheres (Monger and Price, 2002). In Early Jurassic times, the convergence of the North American plate with continuous subduction zones led to the accretion of back-arc basins and offshore arcs that formed continental margins and new continental arcs (Monger and Price, 2002). By the Late Cretaceous, the newly formed Western Cordillera was located near its present-day position (Monger and Price, 2002).

The Western Cordillera can be subdivided into Canadian Cordillera and the Laramide Orogen (USA). In the Canadian Cordillera, the porphyry copper deposits were emplaced post-orogeny accretion from the Late Cretaceous to the Eocene (100 - 33 Ma) (McMillan et al., 1996; Leveille and Stegen, 2012). Conversely, in the Laramide Orogen, their formation is associated with magmatism and compressional deformation between 80 to 40 Ma (McLemore, 2008; Leveille and Stegen, 2012).

The Andean Belt results from the subduction of the Nazca Plate beneath the South American Plate, its formation began in the Middle Jurassic and is still active today (Ramos, 2009). The Andean Belt is subdivided into three sections, each characterized by different geodynamic events. The northern portion represents the major collision phases during the Paleogene, involving the accretion of oceanic crust due to the collision of oceanic terrains (Ramos, 2009). The central portion has experienced more active subduction in the Late Neogene and is characterized by thermal uplift (Ramos, 2009). In the southern part, the uplift is associated with ridge collision, which is a result of the closure of back-arc oceanic basins (Ramos, 2009).





The Western Cordillera and Andean Belt host together the world's largest concentration of porphyry copper deposits, 177 in total (Fig. 4). Among these, the majority have an Eocene-Oligocene age (82 deposits), followed by Miocene to Pliocene (49 deposits) and Paleocene ages (46 deposits). Emphasis is given later to the Andean Belt, as it holds most of the known deposits. The West-Pacific Orogenic Belt, considered as a polycyclic orogeny, has undergone repeated tectonic activity and magmatism since Late Precambrian (Matsumoto, 1977). The subduction of the Pacific Plate beneath the Eurasian continent resulted in the formation of island arcs that migrated toward the Pacific Ocean (Aubouin, 1990; Izosov et al., 2020). These collision events occurred between the Middle Paleogene and Neogene and are responsible for a major regional plate reorganization and contributed to the present-day configuration of the magmatic arcs (Hedenquist et al., 2012). The development of porphyry copper deposits within the West-Pacific Orogenic Belt is associated with the reorganization of tectonic plates during the early to middle Miocene (Hedenquist et al., 2012).

The West-Pacific Orogenic Belt encompasses a total of 66 known porphyry copper deposits, with the majority being of Miocene to Pliocene age (55 deposits). The remaining 11 deposits of Eocene-Oligocene ages are principally found in Papua New Guinea and in the Philippines.

The development of the Alpine-Himalayan Orogeny results from a transpressive tectonic regime where the African, Arabian, and Indian tectonic plates underthrust the Eurasian continental plate (Storetvedt, 1990). This following collisional process commenced in Late-Cretaceous (Stefanini and Williams-Jones, 1996; Perello et al., 2008), leading to the closure of the Tethys Ocean and to major phases of mountain building, forming the Alpine belt during the Paleocene (Storetvedt, 1990; Handy et al., 2015). In the Asian portion of the orogeny, the Himalayan Belt, rose rapidly between the Middle-Eocene until the Late-Miocene due to the breakoff of the Neo-Tethyan slab (Ji et al., 2020, 2016).

The Alpine-Himalayan Orogeny encompasses a total of 71 known porphyry copper deposits. The two oldest deposits, dating back to the Paleocene, are located in the older section of this orogeny, namely the Alpine Orogeny. This deformation belt also contains a higher number of Eocene to Pleistocene deposits (44 ore bodies) when compared to the ones found on the Himalayan Mountain range (25 ore bodies).

## 3.2 Paleoclimatic evolution

We analysed the paleo-precipitation reconstructions from Valdes et al. (2017) to evaluate variations in precipitation conditions through time for each of the three tectonic regions of interest.

The northern part of the Western Cordillera, where deposits do not exceed 64 Myr (Yanites and Kesler, 2015) of age, initially experienced humid conditions with mean precipitation levels around 5 m/yr (Valdes et al., 2017). The region then gradually becomes drier, with present-day precipitation values reaching ∼3 m/yr (Fig. A1) (Valdes et al., 2017). The central portion of the Cordillera (Laramide Orogeny) started as a dry environment with precipitation rates as low as 0.1 m/yr at the time of the formation of the first deposits, followed by an increase of up to 3 m/yr in recent times (Fig. A1). A similar pattern can be observed for the deposits found in Alaska. In contrast, the southern regions of the Western Cordillera, encompassing Mexico and Central America, have experienced 3 main paleo-climatic conditions. During the early Cenozoic, these regions were characterized by a wet tropical climate with precipitation rates of up to 7 m/yr. From 20 to 10Ma, these regions became




drier, with precipitations of the order of 1 m/yr. For the last 10 Ma, the precipitation increased reaching up to 6 m/year (Fig. A1).

In the central part of the Andean Belt, where the oldest deposits (64 Ma) are concentrated, we note a slow increase in rainfall
since their emplacement, from 1 m/yr to nearly 2 m/yr around 10 Ma (see Fig. A1). From this point forward, the paleo-climate reconstructions show a wetter regime with rainfall reaching up to 3 m/yr at present-day. The younger deposits, in the southern portion of the Andean Belt, experienced decrease in rainfall since their emplacement (22 Ma), with precipitations varying from 2 to 1 m/yr. In the northern Andean Belt, where the deposits are mainly from the Neogene, we note an overall high precipitation regime, where the rainfall fluctuates from 6 m/yr at 20 Ma, to slightly dryer conditions at 5 Ma (4 m/yr), and up to 8 m/yr at
present-day (Fig. A1).

In the Alpine-Himalayan Orogeny, the oldest deposits in the Alps have transitioned from a predominantly dry climate to sporadic episodes of enhanced precipitation. The Eocene to Neogene Alpine deposits have generally remained under relatively dry conditions (1 m/yr at 40 Ma), with the exception of the last 5 Ma, where we note an increase of up to 4 m/yr precipitation (Fig. A1). For the deposits in the Himalayas, which date back no further than 46 Ma, there has been an overall increase in
precipitation regimes that varied from 1 m/yr at 50 Ma to up to 6 m/yr at present-day (Fig. A1).

In the West-Pacific Orogenic Belt, the deposits are no older than 31 Ma. Except for porphyry bodies of Oligocene age, all the younger deposits were emplaced in regions under tropical climatic conditions, where precipitation levels reached up to 8 m/yr and remained under stable precipitation forcing for the past 25 Ma (Fig. A1). As for the older deposits, we can track back their evolution to locations with lower precipitation rates. As an example, the deposits in Papua New Guinea, between 20 to 10 Ma,
experienced fluctuations in precipitation ranging from 1 to 4 m/yr.

### 3.3 Global evaluation of predicted porphyry copper emplacement depths

Based on temporal variations of tectonics and climate forcings, we calculate exhumation rates and estimate emplacement depths for the 314 deposits described in Yanites and Kesler (2015). Our analysis utilises the predicted erosion rates extracted from the simulation performed with *goSPL* (see section 2) (Fig. 5). An emplacement depth of 2 km was assumed in Yanites and Kesler
(2015), combined with known ages of the porphyry deposits, to estimate exhumation rates that were then benchmarked against independent measurements for 30 of these deposits (Singer et al., 2008) . Their findings revealed a mean difference of 0.062 mm/yr. Predicted exhumation rates from landscape evolution simulation for the same deposit sites show a comparable mean difference of 0.04 mm/yr with Singer et al. (2008) measurements (Fig. 5a). These results indicate that the findings obtained from both approaches have the same order of magnitude, indicating a high level of similarity and comparability.

In our approach, exhumation rates and emplacement depths are calculated at the location of each of the 314 deposits as well as their surrounding area (within a 25 km diameter). Measuring the surrounding area mitigates potential distortion arising from low-resolution data (Fig. 5b and c). Mean simulated exhumation rates range from 0.002 to 0.620 km/Myr, with a normal centered distribution around 0.1 km/Myr (Fig. 5b). The distribution pattern of these rates is consistent with the exhumation rates extrapolated by Yanites and Kesler (2015), which were inferred based on the chronological record of the deposits (illustrated
in Fig. 5d). Additionally, a latitudinal analysis of the deposits suggests a correlation between geographic location, reflective of





climatic conditions, and both exhumation rates and emplacement depths (Fig. 5d). Of the 36 deposits with exhumation rates exceeding 0.3 km/Myr, the majority (25) are situated within the southern tropical belt (0-15°S-band). For the remaining ones, none of them are found outside 16°N and 17°S latitudes. Exhumation rates smaller than 0.3 km/Myr are evenly distributed across latitudes (Fig. 5d).

As expected, we find a relation between the ages of deposits and their exhumation rates (Fig. 5e). The older the deposits are, the smaller the rates tend to be, a trend that is aligned with previous findings (Yanites and Kesler, 2015). Also, within similar emplacement depths, older deposits tend to have smaller exhumation rates (Fig. 5e), conformable with literature data (Yanites and Kesler, 2015; Singer et al., 2008). Yet, climate exerts a strong influence on the exhumation patterns. The largest calculated exhumation rate (0.6 km/Myr) is seen in Ecuador, for a deposit of 20 Myr old. This contrasts with a neighboring deposit of 15

Myr old with an exhumation rate of 0.5 km/Myr.

We found that the simulated emplacement depths from the landscape evolution model are typically at 2 km or shallower (Fig. 5c), which is consistent with values proposed in the literature (Richards, 2022; Sillitoe, 2010; Yanites and Kesler, 2015). Although we note that the deepest calculated emplacement depths occur between 0° and 30°S (Fig. 5d), no strong correlation between the distribution in latitudinal bands (indicative of prevailing climate) and calculated emplacement depth is observed, as

expected. While our analysis reveals relevant impact of climate on these calculations, it is important to emphasize that climatic factors alone do not solely dictate these estimations.

## 4    Discussion

### 4.1    Exhumation rate (ER) and emplacement depth (ED)

Approximately 50% of the averaged values for calculated exhumation rate and emplacement depths align with independent

data published in the literature (Yanites and Kesler, 2015; Singer et al., 2008). However, discrepancies arise when comparing individual deposit data.

For instance, the lowest simulated exhumation rates are for deposits in the westernmost part of Pakistan (Reko Diq complex) and Iran, with values ranging from 0.002 to 0.007 km/Myr and ages varying between 11 and 21 Ma. In the literature, the ER values obtained based on the thermal history for the Reko Diq complex range between 0.20 to 0.79 km/Myr (Fu et al., 2005).

Another example is the world's youngest giant porphyry Cu-Au deposit, Ok Tedi in Papua New Guinea (dated at 1.4 Ma, van Dongen et al. (2010)), which has calculated exhumation rates of 0.3 km/Myr. In contrast, the exhumation rate derived from SHRIMP age data is between 3 to 5 km/Myr (van Dongen et al., 2007). It is important to mention that both aforementioned examples are of deposits situated in highly active tectonic regions.

The emplacement depths of deposits like Butte (Rusk et al., 2008) in Montana (USA) and Panguna (Eastoe, 1978) (eastern

Papua New Guinea), obtained from fluid inclusion, vary from 5 to 9 km and 2 to 3 km, respectively. The predicted emplacement depth from the Landscape Evolution Model for Butte deposit is 1.8 km and for Panguna 0.9 km. In section 2, we discussed several assumptions and simplifications from the landscape evolution model. Moreover, the model carries inherent uncertainties




from the initial input data (paleo-elevation and paleo-precipitation) and assumes that all deposits are exposed at the surface and have experienced minimal erosion.

The Andean Belt *per se* concentrates 90 of the studied porphyry copper deposits. Their distribution can be associated with the presence of the morphostructural volcanic zones, which are mainly controlled by the absence of flat-slab subduction for at least the past 12 Myr (Ramos, 2009). The absence of PCDs in some latitudinal bands along the Andean Belt (*e.g.*, between 10° and 13°S - Fig. 6a) coincides with the presence of flat slab subduction in the past 12 Ma to present and the presence of aseismic ridges, like the one formed by the Nazca Plate collision (Ramos, 2009).

Remarkably, nearly all the highest exhumation rates were calculated within the Andean Belt, which is consistent with high precipitation rates described for the region (see section 3.2). Additionally, all inferred emplacement depths exceeding 5 km were exclusively found within this region (Fig. 6a). When contrasting exhumation rate values obtained from Yanites and Kesler (2015) with those derived from our study for the three specific sites depicted in Fig. 6a, the results indicate that our model yields slightly lower exhumation rate values (Fig. 6b and c). Despite this difference in exhumation rate, the projected
emplacement depth for these sites lies between 2 and 3 km (Fig. 6c).

## 4.2   Paleo-elevation

We acknowledge that the paleo-elevation model provided in Scotese and Wright (2018) PALEOMAP dataset is associated with non-quantified uncertainties which can also drastically influence our results. As no other continuous open-source global paleo-elevation model exists for the period of interest, we evaluate differences between Scotese and Wright (2018) and Boschman
(2021) paleo-elevation models in South America. Boschman's study reconstructs the paleo-elevation of the Andes for the last 80 Myr. Their reconstruction is derived from a compilation of paleo-elevation estimates collected from various sources in the literature (*e.g.*, palynology, stable isotopes, thermochronology, fossil).

The comparison between Boschman (2021) and Scotese and Wright (2018) (Fig. A2) shows that the further back in time, the more discrepancies arise, with differences of up to 3000 m in elevation. For deposits located towards the north of 15°S, the
differences tend to decrease in the last 15 Ma (see Fig. A2a and b). South of 15°S (see Fig. A2c), we note that between 20 and 5 Ma, Scotese and Wright elevation values tend to underestimate elevations proposed by Boschman, with differences of nearly 2000 m. Aside from the complex geology in and within these regions as will be discussed in section 4.5, the differences in elevation calculations are attributed to the varying levels of detail in global (Scotese and Wright, 2018) versus regional (Boschman, 2021) models. Considering that the landscape evolution model resolution is 5 km, a difference of 2000 m from
different datasets is not negligible. In addition, changes in elevation can also imply differences in slope, which are likely to impact the resulting erosion rates.

Another method to interpret topography consists in describing its morphological features in the form of wavelengths signals (basic land surface parameters) (Olaya, 2009). The impact of tectonics (*e.g.* mountain building or collapsing) on the landscape can be associate to long wavelengths signals, therefore more likely to be represented in a global reconstruction. However, short
wave lengths are comparable to smaller scale events that shape the relief (*e.g.* canyons or incised channels, Richards et al. (2020)), and those tend to be absent in global reconstructions. An example of oversimplification inherent to the global model





resolution is reflected in the final calculation of the copper province Reko Diq complex in Pakistan. This landscape is shaped by complex tectonic processes, such as collision, post-collision extension and back-arc rifting, resulting in 26 magmatic belts (Zürcher et al., 2019). The exhumation rate derived from our model for this region (0.007 km/Myr) is two order of magnitudes

smaller than that observed in the literature (Fu et al., 2005). This discrepancy can be attributed to the lack of detail in the paleo-elevation database, contributing to a likely inaccuracy in exhumation rate estimate.

In terms of initial elevation, we followed the assumptions proposed in Yanites and Kesler which state that present-day porphyry bodies encountered at the surface suffered negligible erosion since their emplacement. However, blind porphyry copper deposits, *e.g.* the Paleogene Resolution Deposit in Arizona (USA), demonstrate that this assumption is incorrect and not in-

significant for all deposits. The Resolution Deposit was found beneath 2 km of postmineral cover (Hedenquist et al., 2012), and therefore theoretically not at the surface yet. Similarly, other deposits may have been eroded away presenting only the remaining few kilometers of the porphyry body. These mistaken assumptions have implications in the final calculations of exhumation rate. For instance, for the Resolution Deposit, the lack of an extra 2 km sediment cover could reflect in an increased exhumation rate calculation from the model.

**4.3  Paleo-climate data**

The presented exhumation rates and emplacements depths have been calculated considering the paleo-elevation derived from Scotese and Wright and associated global paleo-climate model (HadCM3BL-M2.1aD (Valdes et al., 2020)). The modelling approach presented here depends on data resolution of the paleo-reconstruction models used and differences exist between different simulations (Valdes et al., 2020; Li et al., 2022). We acknowledge that rainfall highly impacts the evolution of

physiography/paleo-elevation, and therefore impacts the presented results.

To better understand the uncertainties associated with the imposed rainfall conditions, we analysed an additional simulation that considers the paleo-climate reconstruction from Li et al. (2022) and compared the results with those obtained using Valdes et al. (2020) simulation (Fig. A3). Overall, Li et al. model shows higher precipitation than in Valdes et al..

For regions where PCDs are present, the difference in precipitation reaches up to 4 m/yr, *e.g.* in the northern portion of South

America (Fig. A3), which results in a mismatch in erosion/deposition rates between -1 to 2 mm/yr (Fig. A4). When taken into account since estimated deposit ages, these variations can lead to discrepancies of up to 1 km of erosion or deposition for every million years.

Rainfall values obtained from global models for specific regions also contrast. For instance, precipitation values for deposits formed in the Laramide Orogeny (*e.g.* Butte deposit) obtained from the global model from Valdes et al. (2020) vary from 0

to 3 m/yr (as seen in section 3.2). However, higher resolution precipitations obtained only for the Laramide Orogeny show values from 0.5 to 1.7 m/yr, between 0 to 40 Ma (Retallack, 2007). As a result, our approach considers a paleo-precipitation value that is doubled compared to higher resolution data (Retallack, 2007). These discrepancies in input data reflect on how the emplacement depths might diverge depending on the chosen climatic forcing. Other regions, such as the Alpine Orogeny, also show discrepancies in precipitation data used in the model (Valdes et al., 2020) and data acquired specifically for the region

(Bruch et al., 2011).



## 4.4 Erodibility

While *goSPL* (Salles et al., 2020) is capable of incorporating spatially variable erodibility values, the data resolution for lithologies in most orogenic belts remains insufficient, especially when exploring deeper time periods. Therefore, our approach accounts for a single erodibility component ($\epsilon$) ($4.0 \times 10^{-7}$/yr) which implies a single and global type of rock coverage.

Even though PCDs are often present in similar tectonic environments and lithologies, considering a single erodibility component is an oversimplification, which, propagated over simulation time leads to significant discrepancies. Moosdorf et al. (2018), for instance, introduces a global erodibility index where six distinct erodibility values are classified at present-day. Their index is determined by the slopes distributed across lithology classes and their comparison to slopes relative to acid plutonics. Essentially, the lower the average slope of a lithological unit compared to that of acid plutonics, the higher the erodibility index.

In the Central Andean Belt, we identify at least four different rock types varying between granitic, rhyolitic, andesitic and volcano-sedimentary (Geological Map of South America - 1: 5,000,000, Gómez et al. (2019)). According to Moosdorf et al. (2018), six erodibility indexes were classified in this region, in contrast to the single one applied in the model. In section 4.1, we show that all the emplacement depths that exceed 5 km occur within the Central Andean Belt. It suggests that the erodibility value ($\epsilon$) applied for the region is probably too high, predicting deeper emplacement depths. Upon availability, adding refined

lithology maps into the simulation could improve some of these identified mismatches.

## 4.5 Volcano-sedimentary re-burial

All the calculated emplacement depths exceeding 5 km are found within the Central Volcanic Zone (Ramos, 2009)($14°$- $27°$S) in the Central Andes, except for one deposit in the Northern Volcanic Zone in Northern Andes (Ramos, 2009). The Central Volcanic Zone is well known for its complex tectonic evolution (Ramos, 2018; Nocquet et al., 2014) associated with hyper-arid

climatic conditions since late Miocene (Nocquet et al., 2014; Stern, 2004; Reich et al., 2009), leading to low sediment supply (Stern, 2004). The crust of the Central Volcanic Zone is older than its surroundings, exceeding 70 km in thickness (Montgomery and Brandon, 2002; Stern, 2004; Portner et al., 2020), and is delimited from the Northern and Southern Andes by volcanic gaps (Nur and Ben-Avraham, 1983). These volcanic gaps are linked with the Peruvian and Pampean flat slabs (Portner et al., 2020), whilst the Central Volcanic Zone is marked by the normal subduction of the Nazca Ridge (Portner et al., 2020), leading to

significant volcanism in the region. Moving southwards from the Pampean flat slab ($27°$ - $33°$S) is the Southern Volcanic Zone ($33°$ - $46°$S), which is geologically similar to the Central Volcanic Zone (Stern, 2004; Ramos, 2009).

The volcanism in the Central Volcanic Zone is marked by a significant volume of lava and ignimbrite erupted since at least the Early Miocene (*c.a.* 20 Ma), and reaching up to $12x10^3$ km$^3$ (Baker and Francis, 1978). Similarly, the Southern Volcanic Zone has had expressive volcanism from late Miocene to early Pliocene, covering up to 40.000 km$^2$ (Hernando et al., 2019).

The re-burial process associated with the pilling of rocks and sediments due to volcanic activity is not accounted for in our study (Salles et al., 2023). Therefore, the model accounts for less sedimentation than what has been deposited, suggesting overestimated emplacement depth estimations. On the other hand, for the deposits younger than 20 Myr in the Pampean flat



slab, where volcanic activity is limited, we find emplacement depths that coincide with the smallest depth calculations for South America (< 2 km, Fig.5 d).

## 5 Conclusions and future work

Understanding the influence of surface processes on porphyry copper deposit preservation is essential for determining how these deposits can be exhumed to mineable depths or be eroded away altogether. Several studies using machine learning attempted to predict the fertility of a porphyry copper province based on geological, metallogenic and statistical analyses (e.g. Keykhay-Hosseinpoor et al., 2020; Diaz-Rodriguez et al., 2021; Nathwani et al., 2022; Jafrasteh et al., 2018; Abbaszadeh et al., 2021), neglecting the strong effects of climate on deposit exhumation and preservation. Here, we have demonstrated how slight changes in climate impact the exhumation of porphyry copper deposits. We aim to inspire future research in the field of machine learning to consider the role of climatic contributions in deposit preservation and secondary supergene enrichment of copper near the surface.

The landscape evolution model proposed in this study is innovative as it is the first to globally incorporate both paleo-elevation and paleo-precipitation reconstructions in assessing exhumation rates and emplacement depths of porphyry copper deposits. This new approach not only provides a comprehensive global understanding of the correlation between climate history and the existence of the major porphyry copper provinces, but also shows how this technique can be more efficiently applied for regional porphyry copper exploration given better spatial resolution in the input datasets.

We have shown that the approach effectively replicates calculations of emplacement depths on a global scale, with approximately 50% of the values falling within the anticipated ranges of emplacement depth. In addition, our approach allows for quantifying the complex relationships between climate, tectonics, physiography and porphyry copper preservation and could be extrapolated as an exploration tool for assessing the probability of undiscovered porphyry copper deposit preservation.

Inherent to any numerical model are simplifications and assumptions. In this case, factors such as erodibility, reburial, initial depth of deposit and surface rock lithologies are recognized as oversimplified. We also highlight how minor variations in dataset can contribute to disparities in final predictions and potentially misleading interpretations, underscoring that uncertainties are influenced by model assumptions and dataset resolution. Due to such simplifications and uncertainties, some of our calculated values for exhumation rates and emplacement depths do not match data found in the literature. Nevertheless, we have established workflows that enable to test and quantify the differences between different paleo-datasets (either paleo-climate or paleo-elevation). By identifying which dataset (paleo-elevation or paleo-rainfall) or simplifications most affect the model, we aim to target future directions for model improvements.





**Figure 1. Comparison between predicted enhanced paleo-elevation and corresponding paleo-elevation map**. **a.** The top panel shows the input elevation conditions for 50 Ma at 0.1° resolution (SW2018 Scotese and Wright (2018)) and the bottom two represent model outputs after the first and second simulation steps (0.05° resolution), highlighting the geomorphological imprints induced by surface processes on the landscape. **b.** Regional scale elevations for the 2 domains highlighted in **a**, with the initial paleo-elevation SW2018 (Scotese and Wright, 2018) on the left, the resulting enhanced paleo-elevation under purely erosive scenario (centre) and after applying the tectonic correction and reaching dynamic equilibrium (right). **c.** Temporal change between imposed tectonic rates from corrected topography and erosion rates at 50 Ma (blue curve). This curve is used to estimate when dynamic equilibrium conditions have been reached. Corresponding continental hypsometric curves for the given paleo-elevation at 50 Ma (purple) and simulated enhanced paleo-elevation (black). Red curve in the inset shows the differences between the two hypsometric curves.



**Figure 2. Global scale landscape evolution model from the Cretaceous to present-day**. Left panels represent simulated elevations for 4 time-slices accounting for surface processes impact and highlighting continental topography and associated river networks (dark blue). Right panels show associated erosion/deposition rates (blue/red respectively) for the considered time slices.





**Figure 3. Reconstructed sediment fluxes and continental sedimentary basin evolution**. **a.** Simulated global average trends in erosion rate, net sediment flux delivered to the ocean and cumulative deposition areas for the Phanerozoic with major orogenic episodes over the past 100 Myr. **b.** Global changes in continental paleo-precipitation (Valdes et al., 2020) and paleo-elevation (Scotese and Wright, 2018) (top panel) and comparison of predicted long-term mean denudation rate with estimates from preserved sediments (WM2007 (Wilkinson and McElroy, 2007)) and recent rates compiled in WCM2013 (Willenbring et al., 2013) **c.** Distribution of the 500[th] largest rivers water and sediment flux at 0 Ma. **d.** Log-log plots representing the distribution of modeled water discharge and sediment flux against basin drainage area (blue and orange circles respectively). Relationship between the two variables is calculated with a power law curve fitting and is represented by the black lines.



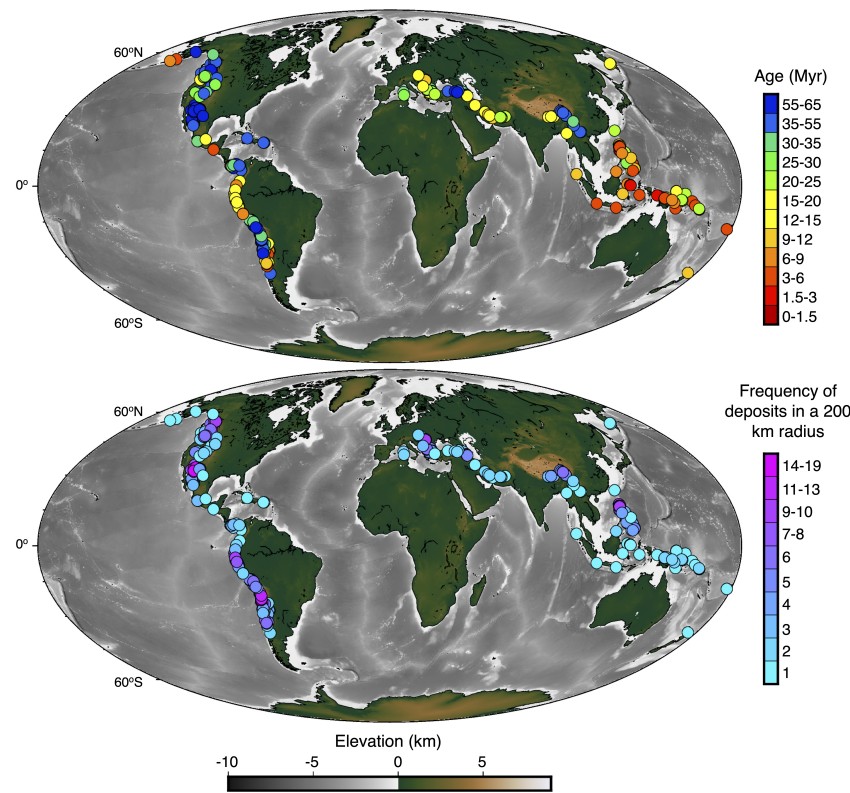

**Figure 4. Distribution of Cenozoic porphyry copper deposit ages and deposit frequency**. The top panel shows the deposit ages with warm colours representing young ages, whereas cool colours represent old ages (Singer et al., 2008). The bottom panel highlights the Cenozoic porphyry copper deposits counted within a 200 km radius of each central deposit. Warm colours represent high frequency, whereas cool colours represent fewer deposits (Yanites and Kesler, 2015).



**Figure 5. Global evaluation of exhumation rates and emplacement depths of porphyry copper deposits. a.** Comparison between exhumation rates using the independent depth estimate for 30 samples (Yanites and Kesler, 2015) and predicted rates assuming a 2.0 km emplacement depth and obtained from the enhanced paleo-elevation model. **b.** Distribution of the predicted mean exhumation rates from the porphyry copper deposits (Singer et al., 2008) assuming a 25 km surrounding area for each site. **c.** Calculated porphyry copper deposits emplacement depths distribution from the mean exhumation rates calculated in **b.** Box and whisker plots of exhumation rate distributions in 15° bands with the associated emplacement depths (**d**) and temporal distributions based on deposits ages (Singer et al., 2008) using 5 Myr bands (**e**).







**Figure 6. Evaluation of South America porphyry copper deposits exhumation histories and emplacement depths**. **a.** South America copper deposits formation ages (Singer et al., 2008; Yanites and Kesler, 2015) and predicted emplacement depths from the exhumation rates modelled with the enhanced paleo-elevation simulation. **b.** Histories of exhumation induced by the surface processes action forced by paleo-precipitation (Valdes et al., 2020) and paleo-elevation (Scotese and Wright, 2018) for porphyry copper deposit sites 1 and 2 (defined in **b**). **c.** Reconstructed positions of site 3 (defined in **a**) over time based on deposit age and the plate reconstruction rotations (PALEOMAP Scotese and Wright (2018)) showing the evolution of erosion and deposition rates (blue and red respectively). Right panel shows box and whisker plots of simulated precipitations (Valdes et al., 2020) and exhumation rates within a 25 km radius around the deposit site.





*Code availability.* The scientific software used in this study, *goSPL* (Salles et al., 2020), is available from https://github.com/Geodels/gospl and the software documentation can be found at https://gospl.readthedocs.io. We also provide a series of Jupyter notebooks used for processing the data sets and model outputs that can be followed to reproduce some of the figures presented in the paper and can be accessed from https://github.com/Geodels/paleoPhysiography and https://github.com/biahadler/porphyry-data.

*Data availability.* The PALEOMAP paleo-elevation reconstruction and related GPlates rotation and geometry files can be downloaded from https://www.earthbyte.org/paleodem-resource-scotese-and-wright-2018/ (last access: 21 May 2024). Paleo-precipitation maps from the HadCM3BL-M2.1aD model (Valdes et al., 2020) are available from the Bristol Research Initiative for the Dynamic Global Environment (BRIDGE) website. All the maps with enhanced paleo-elevation reconstruction (Salles et al., 2022) for the Phanerozoic are available from HydroShare: http://www.hydroshare.org/resource/0106c156507c4861b4cfd404022f9580. The paleo-elevation reconstruction from (Boschman,

2021) is available from https://data.mendeley.com/datasets/h2w7pshz44/1 (last access: 21 May 2024). Palaeoclimate simulations from (Li et al., 2022) are available in the original article, and can be accessed at: https://figshare.com/articles/dataset/A_high-resolution_climate_simulation_dataset_for_the_past_540_million_years/19920662/1 (last access: 21 May 2024). The dataset containing the 314 studied mines in this project can be found within Yanites and Kesler (2015) supplementary information. We provide data as netCDFs files for a few examples of mines that were used to produce Figure A2 and the rotation file utilized in that as well, available at: https://zenodo.org/uploads/11239057.



**Figure A1. Global paleo-precipitations maps over the past 60 Ma**. Darker blues represent higher precipitations, which are usually recurrent over the equatorial region. White patches depict dry regions. The location of each PCD is projected on the maps as from their age of emplacement.





**Figure A2. Mean elevations around 25 km radius from a point over time (left) considering the different data sets: Scotese and Wright (2018) PALEOMAP, *goSPL* Salles et al. (2023), and Boschman (2021)** The whisker plots and the blue line refer to the dataset from PALEOMAP (Scotese and Wright, 2018). The green line shows the elevation values from Boschman (2021) model. The pink line is the mean elevation from the Landscape Evolution Model from Salles et al. (2023). The red line shows the age of emplacement of the porphyry copper deposit related to the depicted location.





**Figure A3. Differences in precipitations over time between paleo-climate models**. Negative values (purple) represent precipitation conditions from Li et al. which are larger than the ones predicted by Valdes et al. (positive ones are shown in green).



**Figure A4. Differences in erosion/sedimentation through time between Valdes et al. and Li et al.**. Negative values represent where the precipitation from Li et al. is larger than the ones considered in Valdes et al., and positive ones the other way around.





## A1

*Author contributions.* B.H.B & T.S. conceived the study and conducted the numerical experiments, B.H.B., T.S. & C.M. analyzed the exhumation rates. B.H.B wrote the initial draft, and all co-authors reviewed the manuscript.

*Competing interests.* The authors declare no competing interests.

*Acknowledgements.* This work was undertaken through STELLAR (Spatio-temporal exploration for resources) industry collaboration supported by BHP. This research was undertaken with the assistance of resources from the National Computational Infrastructure (NCI), which is supported by the Australian Government and from Artemis HPC Grand Challenge supported by Sydney Informatics Hub at the University of Sydney.



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
