# Peer review of "The roles of surface processes on porphyry copper deposits preservation"

_EGUsphere, 2024_

## Author Response (AR2)

**Author's response to the reviews on paper: "The roles of surface processes on porphyry copper deposits preservation"**

We thank both reviewers for their insightful comments and their efforts in helping improve our manuscript. We are particularly grateful that both reviewers found the study valuable and only suggested minor revisions. Referee #1 describes this as **"a unique study"** and notes that **"the results are of considerable interest from the perspective of mineral deposit geology."** Referee #2 highlights **"the use of a sophisticated numerical code of high-resolution"** and suggests useful modifications to clarify the limitations inherent to the model, as well as to provide a more detailed explanation of the main equations and how simplified formulations might affect the interpretation of results. We have addressed these comments in the revised manuscript and believe the resulting version is significantly improved.

In the following, we address the technical points and minor issues raised by the reviewers, outlining the suggestions and the corresponding modifications made to the manuscript.
* * *
**Referee 2 comment i):** In equation (1), the term related to the fluvial processes considers the local precipitation P and the upstream total area A. In line 84, the authors define the water flow as the product P times A. However this is only true if the precipitation P is spatially constant.

The correct flow discharge should be calculated as the integration of the precipitation in each upstream numerical cell. Therefore, the water flux in one point of the mesh depends on the upstream precipitation and not on the local precipitation.

This is specifically important in the eastern side of the Andean Cordillera where orographic precipitation can create significant lateral variations in precipitation rate. Therefore, I think it is important to discuss more about this in the manuscript, and how the adopted simplification can affect the results.

**Response to Comment 1:** We thank the reviewer for their careful observation. We agree that assuming water flux as the product of local precipitation and upstream area is only valid under spatially uniform precipitation—an assumption that does not hold in many real-world settings, particularly where orographic effects lead to strong lateral precipitation gradients, such as the eastern Andes.

In response, we have revised **Equation (1)** to better reflect how goSPL handles spatially variable precipitation across the catchment. In the updated formulation, the flow discharge at each cell is calculated as the **sum of the local rainfall contribution** (local precipitation multiplied by the local area) and the **accumulated upstream discharge**.

This explicitly shows that the water flux at a given mesh point depends not only on local precipitation but also on all upstream precipitation contributions.

The original mass continuity equation was:

$$\frac{\partial z}{\partial t} = U + \kappa \nabla^2 z + \epsilon P^d (PA)^m \nabla z^n$$

where the water flux was simplified as PA, assuming spatially constant precipitation.

In the revised manuscript, we define the local change in elevation over time as:

$$\frac{\partial z_i}{\partial t} = U_i + \kappa \nabla^2 z_i + \epsilon P_i^d Q_i^m \nabla z_i^n$$

where the **local flow discharge** $Q_i$ is expressed as:

$$Q_i = P_i A_i + \sum_{j \in up} Q_j$$

Here, $A_i$ is the area of cell i, $P_i$ is the local precipitation, and the summation is for all nodes j that belong to the upstream set (up). This revised equation makes explicit that the water flux $Q_j$ depends on both the local precipitation and the cumulative discharge from upstream nodes. We have clarified this change in the revised manuscript (lines 79 – 81, and lines 91- 92), highlighting its relevance for regions with significant precipitation gradients.
* * *
**Referee 2 comment ii):** Please, give some description of how the tectonic forcing was added in the model. Give details of how the U term is calculated to simulate the tectonic correction to reach the dynamic equilibrium. The authors referred to Salles et al. (2023), but I think a short description in the paper can contribute to the understanding of the mathematical procedure adopted in this work.

**Response to Comment 2:** Following reviewer's advice, a description of how tectonic forcing U has been incorporated into the model has been added at line 132 of the revised manuscript:

*"The second and final round of simulations are run considering the same initial input files, however with the computed tectonic forcing, added to it (Salles et al., 2023) (Fig. 1). The tectonic forcings is computed from the mismatch between the model predictions and a geologically reconstructed target location. The mismatch indicates that either surface processes parameters (erodibility) need to be tuned, or that other forcings such as dynamic uplift or subsidence need to be considered. If the second is decided, a new tectonic forcing based on the filtered mismatch map which only consider long wave lengths and short amplitudes, is imposed. Mismatch percentages and correlation coefficients guide the calibration of the paleo-elevation maps."*

This addition outlines the calculation of U and how the correlation coefficients and mismatch percentages guide the dynamic equilibrium based on Salles et al. (2023).
* * *
**Referee 2 comment iii):** The paper does not mention any isostatic compensation. Therefore, I assume that the model does not take into account isostasy (local or regional). My question is: how does the vertical displacement of the plate, caused by the load of the cordillera and sedimentary layers in the foreland basins, affect the present results of exhumation rates? The authors briefly commented on this on line 385, discussing about the effect of re-burial, but I think that flexural effects can significant impact the exhumation rate, aspect not explored in the text.

**Response to Comment 3:** We appreciate the reviewer's comment regarding the omission of explicit isostatic compensation (local or regional) in our modelling framework. Indeed, as correctly noted, the current version of goSPL does not directly account for flexural isostasy or lithospheric flexure in response to erosional unloading or sedimentary loading.

However, our data assimilation approach implicitly captures some of these effects by iteratively adjusting the evolving topography to remain consistent with independent paleo-elevation reconstructions (mismatch maps). This means that vertical motions such as regional uplift or subsidence - whether driven by tectonics or flexural loading - can be partially integrated into the model through these boundary conditions, although we cannot, with confidence, ascribe the uplift to a specific driver such as tectonic motion or flexural loading.

That said, we agree that flexural responses to sediment loading in foreland basins and unloading in orogenic regions can strongly influence exhumation rates, especially over $>10^5$-year timescales and regional scales of hundreds of kilometres. These feedbacks are not fully captured in our current model setup, and therefore, the exhumation rates presented here may underestimate or misrepresent areas where flexural effects dominate the vertical signal.

In future work, coupling goSPL with a flexural isostasy module or using dynamic lithosphere models would improve the treatment of these processes and refine estimates of both burial and exhumation in foreland settings. We clarify this limitation in the revised manuscript in lines 70 to 74:

"*Flexural responses due to erosional unloading and depositional loading influence long-term drainage evolution, landscape rejuvenation, and sedimentation, reflecting lithospheric strength and mantle buoyancy (Sacek, 2014). While goSPL does not explicitly model these processes, the assimilation method implicitly accounts for them by aligning modelled elevations with paleo-elevation reconstructions (Salles et al., 2023). As an*

*example, the approach captures subsidence patterns near large deltaic systems (e.g., Amazon, Bengal fans, Pelotas Basin)."*

And expand the discussion accordingly in lines 401 to 405:

*"Re-burial processes can also initiate flexural responses, which are indirectly accounted for in the model. These processes are particularly significant in the Northern Andes, where the interplay between flexural isostasy and surface processes helps explain the drainage reversal of the Amazon River (Sacek, 2014). The isostatic effect of volcanic loading can induce local subsidence, which if not taken into account, may lead to overestimation of the final exhumation rates. Conversely, neglecting flexural isostasy associated with sediment unloading can result in an underestimation of exhumation rates."*

**Additional alterations:**

**Errata:** Figure 2 was incorrectly referenced in section "Paleo-elevation and paleo-precipitation forcings", the correct figure to be cited is Figure 3. The alterations have been made to the manuscript accordingly.

**Line 397:** We have altered the initial text from "The re-burial process associated with the pilling of rocks and sediments due to volcanic activity is not accounted for in our study." To "The re-burial process associated with the overlaying of volcaniclastic rock is not accounted for in our study" for enhanced readability.